# Evaluation of Class IIa Histone Deacetylases Expression and In Vivo Epigenetic Imaging in a Transgenic Mouse Model of Alzheimer’s Disease

**DOI:** 10.3390/ijms22168633

**Published:** 2021-08-11

**Authors:** Yi-An Chen, Cheng-Hsiu Lu, Chien-Chih Ke, Sain-Jhih Chiu, Chi-Wei Chang, Bang-Hung Yang, Juri G. Gelovani, Ren-Shyan Liu

**Affiliations:** 1Institute of Clinical Medicine, National Yang Ming Chiao Tung University, Taipei 112, Taiwan; yachen0414@gmail.com; 2Molecular and Genetic Imaging Core/Taiwan Mouse Clinic, National Comprehensive Mouse Phenotyping and Drug Testing Center, Taipei 112, Taiwan; rocket2350@yahoo.com.tw (C.-H.L.); e2689@ym.edu.tw (S.-J.C.); 3Department of Nuclear Medicine, Cheng Hsin Hospital, Taipei 112, Taiwan; 4Industrial Ph.D Program of Biomedical Science and Engineering, National Yang Ming Chiao Tung University, Taipei 112, Taiwan; 5Department of Medical Imaging and Radiological Sciences, Kaohsiung Medical University, Kaohsiung 807, Taiwan; 6Department of Medical Research, Kaohsiung Medical University Hospital, Kaohsiung 807, Taiwan; 7Drug Development and Value Creation Research Center, Kaohsiung Medical University, Kaohsiung 807, Taiwan; 8National PET and Cyclotron Center (NPCC), Department of Nuclear Medicine, Taipei Veterans General Hospital, Taipei 112, Taiwan; cwchang@vghtpe.gov.tw (C.-W.C.); bhyang@vghtpe.gov.tw (B.-H.Y.); 9Department of Biomedical Imaging and Radiological Sciences, National Yang Ming Chiao Tung University, Taipei 112, Taiwan; 10Office of the Provost, United Arab Emirates University, Al Ain P.O. Box 1551, United Arab Emirates; jgelovani@uaeu.ac.ae

**Keywords:** Alzheimer’s disease, epigenetic regulation, histone deacetylase, amyloid-β, PET imaging, HDAC inhibitor

## Abstract

Epigenetic regulation by histone deacetylase (HDAC) is associated with synaptic plasticity and memory formation, and its aberrant expression has been linked to cognitive disorders, including Alzheimer’s disease (AD). This study aimed to investigate the role of class IIa HDAC expression in AD and monitor it in vivo using a novel radiotracer, 6-(tri-fluoroacetamido)-1-hexanoicanilide ([^18^F]TFAHA). A human neural cell culture model with familial AD (FAD) mutations was established and used for in vitro assays. Positron emission tomography (PET) imaging with [^18^F]TFAHA was performed in a 3xTg AD mouse model for in vivo evaluation. The results showed a significant increase in HDAC4 expression in response to amyloid-β (Aβ) deposition in the cell model. Moreover, treatment with an HDAC4 selective inhibitor significantly upregulated the expression of neuronal memory-/synaptic plasticity-related genes. In [^18^F]TFAHA-PET imaging, whole brain or regional uptake was significantly higher in 3xTg AD mice compared with WT mice at 8 and 11 months of age. Our study demonstrated a correlation between class IIa HDACs and Aβs, the therapeutic benefit of a selective inhibitor, and the potential of using [^18^F]TFAHA as an epigenetic radiotracer for AD, which might facilitate the development of AD-related neuroimaging approaches and therapies.

## 1. Introduction

The characteristics of Alzheimer’s disease (AD) include abnormal deposition of amyloid-β (Aβ) plaques, synaptic degeneration, and neuronal loss [1,2]. Since Aβ is considered a key contributor to the pathophysiology of AD, various therapeutic approaches targeting Aβ have been developed through either β/γ-secretase inhibition or immunotherapy [2,3]. However, the cognitive functions of patients are not improved effectively, suggesting that more than one physiological pathway is involved in AD pathogenesis [2,4]. An increasing number of studies indicate that impaired learning and deterioration of memory are associated with abnormal epigenetic modifications [5,6,7]. The widespread downregulation in gene expression is affected by post-translational histone modifications in which histone deacetylases (HDACs), histone acetyltransferases (HATs), histone methyltransferases (HMTs), and various DNA demethylases participate [8].

Among all HDAC family members, class I HDACs (HDAC1, 2, 3, and 8) are known to be ubiquitously expressed in different cell types and contain a deacetylase domain that can remove an acetyl group from other proteins efficiently. By contrast, class IIa HDACs (HDAC 4, 5, 7, and 9) have tissue-specific expression, primarily in the heart, skeletal muscle, and brain, and exhibit very low intrinsic deacetylase activity [9,10]. Generally, class IIa HDACs directly recruit and inactivate various transcriptional factors or cofactors by a large N-terminal regulatory domain, resulting in transcriptional repression. Numerous reports have indicated that class IIa HDACs are involved in brain development, neurological functions, and neurodegenerative diseases [11,12]. For instance, HDAC4 plays a very important role in neurobiology, as its homeostasis has been shown to be associated with the regulation of the transcription program of synaptic plasticity and with memory [13]. In addition, reports on differential expression have demonstrated that the HDAC4 level is markedly increased in the brain of AD patients and of various AD mouse models [14,15,16]. Several studies have shown that pan-HDAC inhibitors, such as valproic acid (VPA), suberoylanilide hydroxamic acid (SAHA), and sodium butyrate (NaB) significantly restore cognitive performance in neurodegenerating mouse models [12,17]. However, no studies to date have been reported regarding the strategy of targeting class IIa HDACs for AD therapy. Thus, the role of class IIa HDAC in AD and the effect of class IIa HDAC selective inhibitor need to be further evaluated.

Positron emission tomography (PET) imaging with radiotracers targeting AD-related proteins (e.g., Aβ_42_ and phosphorylated tau) provides an excellent non-invasive method for measuring AD pathophysiology [18]. However, the density of Aβ plaque in the brain assessed by PET does not correlate well with neurodegeneration and cognitive dysfunction from clinical observations [19,20]. Various radiotracers developed for imaging neuroinflammation, cholinergic receptors, as well as epigenetic status might help depict AD progression more comprehensively. The molecule 6-(tri-fluoroacetamido)-1-hexanoicanilide ([^18^F]TFAHA), a second-generation imaging probe of class IIa HDACs, particularly HDACs 4 and 5, was developed for the imaging of epigenetic regulation in the brain [21]. Also, [^18^F]TFAHA PET imaging was able to quantitatively assess class IIa HDACs expression and activity in intracerebral glioma xenografts of 9L and U87-MG cells in rats [22]. In this study, we aimed to evaluate the class IIa HDACs expression and in vivo epigenetic status in an AD mouse model by [^18^F]TFAHA PET imaging.

We generated a human neural cell culture model of AD that mimicked AD pathology by overexpressing the human amyloid precursor protein with both Swedish (K670N/M671L) and London (V717I) FAD mutations in the SH-SY5Y cell line, referred to as FAD cells, to investigate the correlation between class IIa HDACs and Aβs. Besides, the therapeutic effect of selective inhibitors was evaluated by the expression of neuronal memory-/synaptic plasticity-related genes. [^18^F]TFAHA microPET imaging was used to monitor class IIa HDACs expression in an AD transgenic mouse model. In vitro results showed that FAD cells exhibited a significant increase in HDAC4 expression, and HDAC4 selective inhibitor treatment upregulated several memory- and synaptic plasticity-related genes. Furthermore, the quantitative results of PET imaging showed that the uptake of [^18^F]TFAHA in whole brains and different brain regions was significantly higher in AD transgenic mice, demonstrating a high class IIa HDACs expression in AD brain.

## 2. Results

### 2.1. In Vitro Characterization of Class IIa HDACs Expression in a Cell Model of AD

To investigate the correlation between class IIa HDACs and Aβ, human neuroblastoma SH-SY5Y cells were exposed to Aβ oligomers that are widely believed to be the most neurotoxic species [23,24]. The aggregation of Aβ peptides was structurally characterized by transmission electron microscopy (TEM), and their toxicity to cells was assayed by cell viability assay (Appendix A). We confirmed that Aβ oligomers were more toxic to SH-SY5Y cells in a concentration-dependent manner than unaggregated Aβs. Immunofluorescent (IF) staining revealed that exposure of Aβ oligomers increased the expression of HDAC4 but not of HDAC5 and HDAC1 in SH-SY5Y cells (Figure 1A–C). This prompted us to further investigate HDAC4 expression following neuronal differentiation.

To further examine whether endogenous neurotoxic Aβs induced the aberrant higher expression of class IIa HDACs, the FAD human neural cell culture model carrying human APP with double FAD mutations was used. Aβ expression and Tau phosphorylation were obviously observed in FAD cells (Appendix A). After induction of neuron differentiation for 7 days, FAD cells, but not WT cells, exhibited an abundance of soluble Aβ species with *M*_W_ up to 90 kDa, which are generally considered neurotoxic oligomers (Figure 2A). Next, IF staining and western blot analysis showed that nuclear HDAC4 levels were increased in FAD cells during neuronal differentiation compared to WT cells (Figure 2B,D). The expression of HDAC5 in FAD cells was found decreased in both differentiated FAD cells and WT cells, with no significant difference. By contrast, the levels of HDAC1 remained unaffected after neuron differentiation. In addition, the level of the *HDAC4* gene was significantly higher in differentiated FAD cells than that in WT cells (Figure 2C). Collectively, the in vitro results showed that the expression of class IIa HDACs, mainly HDAC4, was elevated in the cell model of AD, suggesting that increased HDAC4 levels may correspond to increased neurotoxic Aβs levels.

### 2.2. HDAC4 Selective Inhibitor Treatment UpRegulated Memory- and Synaptic Plasticity-Related Genes

Genes involved in synaptic plasticity were reported to be downregulated in AD. These include *BDNF* (brain-derived neurotrophic factor) and *HOMER1* (homer protein homolog 1) and genes implicated in regulating synaptic function, such as *GLUR* (glutamate receptor subunits), *SYP* (synaptophysin), *SYN2* (synapsin II), and *LGI1* (leucine-rich glioma inactivated 1) [17,25]. Similar results were observed in our FAD cell model (Appendix A). Given that the level of HDAC4 was significantly elevated in our FAD cell model, we next asked whether inhibition of HDAC4 restored the expression of these genes. Tasquinimod (TasQ) is a highly selective inhibitor of HDAC4 and also a clinically tested oral antiangiogenic agent against castration-resistant prostate cancer [26]. To evaluate the effect of TasQ on neuronal cells in AD, differentiated FAD cells were treated with or without TasQ. The result showed that cell growth was inhibited by TasQ treatment in a dose-dependent manner, and the IC_50_ was 243µM for SH-SY5Y-FAD cells (Appendix A). Further, we incubated FAD cells with TasQ at various concentrations ranging from 0 to 50 µM for 48 h. Cell morphology did not significantly change at lower concentrations until the dose was increased to 100 µM (Appendix A). Quantitative RT-PCR analysis showed that 30 µM of TasQ was sufficient to upregulate the level of several plasticity-related genes that have been identified as HDAC4 targets, such as *HOMER1*, *LGI1*, and *SYN2*; this upregulation was concentration-dependent (Figure 3) [13]. Besides, almost of all pan-inhibitor treatments can upregulate *GLUR2*, which encodes the subunit protein of the AMPA receptor: thus, we used this gene as a positive control. These results demonstrated that HDAC4 inhibition upregulated the expression of target genes, suggesting the HDAC4 restored AD-associated gene expression deficits.

### 2.3. In Vivo Epigenetic Imaging Using [^18^F]TFAHA in 3xTg AD and Wild-Type Mice

Previous studies reported that abnormal expression of class IIa HDACs is closely associated with brain dysfunction. Our in vitro results further showed a correlation between HDAC4 and Aβs and also demonstrated the therapeutic benefit of the HDAC4 inhibition. Thus, knowing the expression status of HDAC class IIa may be of great value for further treatment with HDAC inhibitors. In this regard, [^18^F]TFAHA PET imaging was performed to non-invasively detect and evaluate HDAC class IIa expression in AD transgenic mice. The 3xTg AD mouse model, which exhibits not only progressive Aβ deposition but also age-related changes in neuropathologies, is suitable to observe changes in class IIa HDACs expression [27]. We first confirmed that Aβ deposition detected by [^11^C]PiB-PET was higher in 3xTg AD mice than in age-matched WT mice (Appendix A). After [^18^F]TFAHA injection via the tail vein, microPET imaging was performed with both dynamic and static scans. The result of dynamic imaging showed that [^18^F]TFAHA had the highest accumulation in the heart, a relatively low distribution in the brain, and the lowest levels in the muscle during the observation time from 5 to 60 min post-injection (Appendix A). Static images were acquired at 20 min post-injection, and the built-in digital mouse brain atlas was used for alignment and identification of specific brain regions. A visual interpretation of [^18^F]TFAHA-PET showed increased accumulation of [^18^F]TFAHA in whole brains of 3xTg AD mice compared with those of age-matched WT mice (Figure 4). Furthermore, an obvious increased [^18^F]TFAHA signal was observed globally in the cerebrum of 11-month-old 3xTg AD mice (Figure 4B, left panel). These results suggested that [^18^F]TFAHA-PET imaging was able to differentiate AD mice from WT control.

The uptake of [^18^F]TFAHA in the whole brain and in each brain region was measured according to a standard atlas and is shown as SUV (%ID/c.c) (Figure 4C). The uptake in the whole brain of 3xTg mice at 8 and 11 months of age was about 1.15- and 1.63-fold higher than that in age-matched WT mice, respectively (Figure 5 and Appendix A). Besides, 11-month-old 3xTg AD mice exhibited significantly higher [^18^F]TFAHA uptake in most areas examined, including striatum (t = 7.07, *p* = 0.0002), cortex (t = 6.96, *p* = 0.00059), hippocampus (t = 7.8, *p* = 0.0001), basal forebrain (t = 5.96, *p* = 0.00059), thalamus (t = 4.36, *p* = 0.0224), hypothalamus (t = 4.98, *p* = 0.0016), amygdala (t = 8.56, *p* = 0.00006), and cerebellum (t = 5.61, *p* = 0.00081) compared to WT mice. There was no significant difference in [^18^F]TFAHA uptake between 8-month-old and 11-month-old WT mice.

## 3. Discussion

In this study, we demonstrated the correlation between class IIa HDAC and Aβs in Alzheimer’s disease brain by expression analysis and HDAC4 inhibition. Furthermore, we tested the performance of a class IIa HDACs selective radiotracer, [^18^F]TFAHA, for monitoring class IIa HDACs expression and differentiating AD transgenic mice from age-matched controls.

Pathogenic Aβs trigger aberrant epigenetic modifications, such as acetylation of histones, DNA methylation status, and expression of non-coding RNA, contributing to the overexpression of AD-associated genes such as *APP*, *PSEN1*, *PSEN2*, and *BACE*, or to the decreased expression of Aβ-degrading proteases. Thereby, a vicious cycle is established and eventually leads to substantial loss of neurons and synapses and to neurodegeneration [28,29]. However, the role of all class IIa HDACs in AD pathogenesis is not well-defined. Considering the nucleocytoplasmic shuttling properties of class IIa HDACs, they exert different biological effects, highly dependent on their subcellular localization [30]. Our data showed that cytoplasmic HDAC4 was increased after treatment of Aβ oligomers for 48 h (Figure 1). By contrast, the FAD human neural cell model exhibited increased expression of HDAC4 in the nucleus rather than in the cytoplasm after the induction of neuronal differentiation (Figure 2). It is suggested that the level of cytoplasmic HDAC4 was elevated to protect neurons from the short-term neurotoxic insult of exogeneous Aβ oligomers, while the accumulation of nuclear HDAC4 in Aβ-overexpressing cells was involved in pathogenesis mechanisms. These results are in accordance with reports that described HDAC4 as a two-edged sword: neuroprotection in the cytoplasm but neurotoxicity in the nucleus [31,32,33].

The N-terminal domain of HDAC4 is a highly conserved glutamine-rich sequence that has been widely observed to increase the formation of aggregates with other proteins in human neurodegenerative diseases, such as Lewy Bodies, α-synuclein, and neuronal intranuclear inclusion disease (NIIND) [34,35]. Hdac4 knock-down in a mouse model of Huntington’s disease was found to delay cytoplasmic aggregates formation, restore *BDNF* transcript levels, and rescue neuronal and synaptic functions [36]. Whether HDAC4 participates in Aβ aggregation to form fibrils or in hyperphosphorylated tau aggregation to form neurofibrillary tangles is worth further investigation. All class IIa HDACs are known to contain a transcription factor interacting domain that can bind myocyte enhancer factor 2 (MEF2) family members. When HDAC4 enters the nucleus and interacts with MEF2, MEF2-dependent genes implicated in the regulation of cell survival are inhibited, resulting in neuronal apoptosis [37,38]. Moreover, elevated nuclear HDAC4 was observed in the brains of AD patients and in hippocampal pyramidal neurons of various AD mouse models [14,39]. There is emerging evidence that cultured cortical neurons overexpressing a constitutively nuclear HDAC4 mutant downregulate a group of genes essential for synaptic function [13,40]. Thus, the reduction of nuclear HDAC4 in AD represents a strategy for rescuing neuronal and synaptic functions.

HDAC5, an HDAC4 homolog with high similar sequence, has been demonstrated to play a functional role in regulation of cell growth by interacting with MEF2 to silence MFE2-dependent gene transcription programs in cerebellar granule neuron [41]. Kim et al. reported that no learning and memory impairments are observed in Hdac5 KO mice, whereas conditional brain-specific Hdac4 KO mice display significant impairments in learning and memory and long-term synaptic plasticity [42]. Interestingly, Agis-Balboa et al. indicted that the loss of HDAC5 impairs the consolidation of context- and tone-dependent fear memory. Notably, Aβ pathogenesis is mildly affected in HDAC5-deficient transgenic AD mice [43]. Thus, the use of HDAC5-targeting inhibitors as a therapeutic approach for AD is not recommended. Additionally, quantitative results regarding HDACs levels showed that HDAC5 is increased in human AD frontal cortex compared to that in age-matched controls, while the HDAC5 level did not show a significant change in an AD mouse model [15]. Overall, the role of HDAC5 in AD pathogenesis appears to be not well clarified and needs more investigation.

Our results showed that certain HDAC4 target genes involved in synaptic plasticity in FAD cells were upregulated after treatment with TasQ (Figure 3). In fact, TasQ acts as a small molecular oral inhibitor that has entered phase III clinical trials for the treatment of metastatic castration-resistant prostate cancer and has been found to inhibit angiogenesis and tumor growth as well as modulate immune responses [44]. The antitumor mechanism of TasQ is based on allosteric binding to the regulatory zinc-binding domain of HDAC4, thus preventing the formation of the HDAC4/HDAC3/nuclear co-receptor (NCoR) repressor complex. Subsequently, HDAC-mediated deacetylation of histones is inhibited, and the expressions of several HDAC4 client transcription factors, such as hypoxia inducible factor-1α (HIF-1α), is also suppressed [45]. HIF-1α can enhance Aβ generation via promoting β/γ-secretases and inhibiting α secretase. Emerging evidence has also shown that HIF-1α could be a potential therapeutic target for neurodegenerative diseases [46]. Furthermore, HIF-1α has been reported to resist Aβ-derived neurotoxicity, inhibit tau hyperphosphorylation, and cause microglial activation. Thus, it is necessary to investigate the treatment efficacy of TasQ in AD transgenic mice in terms of behavior and neuronal function. On the other hand, the effect of TasQ on HDAC4-regulated memory/synaptic genes was explored at concentrations ranging from 10 µM to 50 µM (Figure 3 and Appendix A); no changes in cell morphology and viability were observed (Appendix A). Recent literature has shown that TasQ is effective in the modulation of HDAC4 at the concentration of 15 µM in Parkinson’s disease patient iPSC-derived dopamine neurons [47]. Although it would be better to assess the off-target effects induced by the higher concentration to support its specificity, the use of TasQ in our study highlighted that HDAC4 is an important molecule and has functional influences.

Given the differential regulation of gene expressions by class-selective HDAC inhibitors, we compared the effects of a class I/IIb HDAC inhibitor, SAHA, a generally considered class II HDAC inhibitor, MC1568, a class IIa HDAC inhibitor, TMP269, a selective HDAC4/5 inhibitor, LMK235, and TasQ on the expression of several genes essential for memory formation and synaptic function. Koppel et al. reported that the inhibition of class II HDACs rapidly increases *BDNF* exon IV as well as *c-fos* and *Arc*, immediate early genes that are associated with synaptic plasticity and memory formation (within 3 h) [48]. In contrast, class I-selective inhibitors exhibited an apparent delay in the induction of these three genes. *GRN*, encoding for progranulin, has been shown to be upregulated by SAHA, whereas no effect on *GRN* expression level was detected using the class IIa HDAC inhibitor TMP269 [49]. The expression of *BDNF-IV*, *c-FOS*, *ARC*, and *PRKCB* genes was significantly downregulated in differentiated FAD cells compared to WT cells (see vehicle-treated group, Appendix A). After HDAC inhibitor treatments, gene expression levels were increased compared to vehicle-treated groups (DMSO). Despite the absence of a significant difference in *GRN* gene level between differentiated FAD cells and WT cells, the expression in FAD cells was markedly upregulated after treatment with SAHA. A time-course analysis revealed that treatment with SAHA and TasQ resulted in a gradual increase of *BDNF*, *c-FOS*, and *ARC* gene levels by 24 h. In contrast, the levels of these genes were increased between 1 and 6 h of treatment with MC1568, dropping at 24 h (Appendix A). Treatments with TasQ, MC1568, and TMP269 did not increase *GRN* gene expression in FAD cells as expected, while SAHA and LMK235 were effective at increasing *GRN* gene expression. Since LMK235 was found to inhibit HDAC4 and HDAC5 activity at a low nanomolar concentration, the upregulation of *GRN* gene expression in this experiment might have been caused by different biological mechanisms. As a control, treatment with all five inhibitors, especially TMP269, reversed the reduction in expression of *PRKCB*, a known HDAC4-regulated gene, in differentiated FAD cells. Of note, a recent study showed that the effect of MC1568 on the inhibition of class IIa HDAC is paradoxical, which might be attributed to the commercially available synthetic isomer or the different substrates used in enzymatic activities [50]. Taken together, the results suggest that these class-selective HDAC inhibitors have differential effects on gene expression.

Before [^18^F]TFAHA was developed, other radiotracers were applied to visualize HDAC activity, including [^18^F]SAHA, [^18^F]FAHA, [^18^F]Bavarostat, and [^64^Cu]CUDC-101 [51]. [^18^F]SAHA is an analog of the most clinically relevant HDAC inhibitor, SAHA and targets class I and IIb HDACs. However, its lower CNS penetration ability limits its application for the diagnosis of brain diseases [52]. [^18^F]FAHA is the first developed radiotracer for PET imaging of class IIa HDAC expression with potent BBB permeability [21]. [^18^F]Bavarostat has been used in rodent and nonhuman primate imaging experiments and showed higher selectivity for HDAC6, a HDAC class IIb enzyme [53]. Aside from fluorine-18 isotopes, [^64^Cu]CUDC-101 was also used for visualizing HDACs in breast cancer, but demonstrated poor accumulation in the brain [54]. A previous study has demonstrated that [^18^F]TFAHA exhibits much higher selectivity for class IIa HDACs in comparison with [^18^F]FAHA; therefore, [^18^F]TFAHA should be suitable for the imaging of epigenetic dysregulation of AD.

[^18^F]TFAHA, a class IIa HDACs-selective radiotracer, was developed to increase the number of fluorine atoms in the acetyl moiety of [^18^F]FAHA and thus has better substrate selectivity. Inside the cells, [^18^F]TFAHA is cleaved by class IIa HDAC, followed by the release of the radiolabeled group [^18^F]trifluoroacetate. In an acute lethality study in mice, trifluoroacetate was categorized as slightly toxic, and mice death was caused by the administration of high doses (>2000 mg/kg) rather than by its metabolites [55]. Another report indicated that the effect of trifluoroacetate on cellular metabolism was due to the decreased concentration of the coenzyme NADPH and glutathione in the liver after intraperitoneal administration of 2000 mg/kg (the LD_50_ value is 1200 mg/kg for mice). However, these coenzyme concentrations returned to normal levels by 24 h [56]. In addition, no histological changes in bone marrow, small intestine, heart, liver, and kidney were observed in long-term experiments using rodent models. As a radiopharmaceutical for PET imaging in this study, the administered dose of 320 MBq/kg of [^18^F]TFAHA, equivalent to 0.5 ng/kg could hardly induce any biological response and possible toxic side effect. Therefore, despite the abundant accumulation of [^18^F]TFAHA in certain organs during PET imaging, its potential toxic side effects are not a concern.

In vivo imaging of class IIa HDACs could reveal temporal associations between epigenetic dysregulation and cognitive decline or amyloid pathology. However, the spatial resolution of microPET imaging in mouse brain is still limited to accurately establish the precise location of [^18^F]TFAHA binding sites in different brain structures such as the nucleus accumbens, the periaqueductal gray, and the dentate gyrus. This might be the reason why we could not observe a high accumulation of [^18^F]TFAHA-derived radioactivity in hippocampus, amygdala, and cerebellum, where HDACs 4 and 5 are abundantly expressed, compared to other brain regions [21]. To compensate for this limitation, autoradiography or the selection of rat models of AD may be helpful for a more detailed understanding of epigenetics in AD pathology.

## 4. Materials and Methods

### 4.1. Cell Culture

The human neuroblastoma cell line SH-SY5Y was kindly provided by Prof. Irene Han-Juo Cheng (National Yang Ming Chiao Tung University, Taiwan). SH-SY5Y cells were cultured in DMEM/F12 medium supplemented with 10% FBS, 100 units/mL penicillin, and 100 μg/mL streptomycin at 37 °C in a 5% CO_2_-containing atmosphere.

### 4.2. Establishment of the FAD Human Neural Cell Culture Model and Relevant Assays

Human amyloid precursor proteins with both K670N/M671L (Swedish) and V717I (London, UK) FAD mutations were overexpressed in SH-SY5Y cells using a lentivirus (see Appendix A). We grew these cells using a Matrigel culture model as described previously [57]. For subsequent immunostaining (IF) and biochemistry analysis, thin-layer (100–300 μm) and thick-layer (about 4 mm) culture models were set up, respectively. Briefly, the cells were pre-differentiated with 5μM retinoic acid (RA, Sigma-Aldrich, St. Louis, MO, USA) for 1 week and then mixed 1:1 with pre-chilled Matrigel (cat. 354234, Corning, Tewksbury, MA, USA) for neural differentiation. The culture medium was replaced twice a week. After dispensing mixtures of pre-differentiated cell/Matrigel (ratio 1:1) into each tissue culture insert of 24-well plates at the density of 5 × 10^4^ cells/well for thick-layer cultures, differentiated FAD cells were then maintained for long-term differentiation with a combination of RA (5μM) and BDNF (50 ng/mL). The mixtures of pre-differentiated cell/Matrigel (ratio 1:10) were seeded onto cover glass plates for IF and microscopy analysis.

### 4.3. Preparation of Aβ Oligomers

One mg of Aβ_42_ peptides (AnaSpec, Fremont, CA, USA) was monomerized by 0.22 mL of hexafluoroisopropanol (HFIP) following the manual’s procedure [58]. In brief, oligomers were grown with the addition of phenol red-free cell culture medium and then incubated at 37 °C overnight. The aggregation state of Aβ_42_ was characterized by structural analysis through TEM, and the neurotoxicity of Aβ oligomers was evaluated by cell viability assays.

### 4.4. Reverse Transcription and Quantitative Real-Time Polymerase Chain Reaction (PCR)

To further investigate the effects of Tasquinimod (TasQ, ABR-215050, Medchemexpress, NJ, USA) on neuronal memory- and synaptic plasticity-related genes, differentiated FAD cells were treated with TasQ at doses between 10 and 50 µM for 48 h. Total RNA was extracted using TRIzol reagent (Invitrogen, Carlsbad, CA, USA) followed by reverse transcription and PCR amplification using the SuperScript III First-Strand Synthesis System (Thermo Fisher Scientific Inc., Waltham, MA, USA). Quantitative real-time PCR was performed by a StepOne^TM^ Real-Time PCR system according to the manufacturer’s recommendations (Thermo Fisher Scientific Inc., Waltham, MA, USA). The following primers were used in this study: HDAC4 F, 5′-GTGGTAGAGCTGGTCTTCAAGG -3′; HDAC4 R, 5′-GACCACAGCAAAGCCATTC-3′; HDAC1 F, 5′-CGGTGCTGGACATATGAGAC-3′; HDAC1 R, 5′-TGGTCCAAAGTATTCAAAGTAGTCA-3′. The specific primers for memory-/synaptic plasticity-related genes are listed in Appendix A. The average threshold cycle (Ct) for each gene was normalized based on the Ct of β-actin or GAPDH.

### 4.5. Western Blot

Cell lysates were prepared with RIPA buffer supplemented with a protease inhibitor cocktail, and equal amounts of protein (30 μg) samples were separated in SDS-PAGE gels. For immunoblotting, the membranes were probed with primary antibodies for HDAC1 (Cat. ab19845, Abcam, Cambridge, UK), HDAC4 (Cat. Ab79521, Abcam, Cambridge, UK), and GAPDH (Cat. ARG10112, Arigo, Taiwan) overnight at 4 °C and then incubated with horseradish peroxidase HRP-conjugated goat anti-rabbit IgG (Cat. ab6721, Abcam, Cambridge, UK). Protein signals were visualized using a chemiluminescent HRP substrate detection system (BioRad, Hercules, CA, USA) and acquired by a luminescence imaging system (UVP BioSpectrum 600, Thermo Fisher Scientific Inc., Waltham, MA, USA).

### 4.6. Transgenic Mouse Model

We used 3xTg-AD (JAX-34830) mice and age-matched control (WT) with the same B6;129 genetic background in this study. The protocol was approved by the Committee on the Ethics of Animal Experiments of the National Yang Ming Chiao Tung University (IACUC number: 1070105, permission date: 12 January 2018). All mice were group-housed in individually ventilated cages (IVC) and had unlimited access to food and water. The room was maintained at a temperature of 20 to 21 °C and relative humidity of 50% to 70% with a 12 h light/dark cycle. Environmental enrichment was provided as a standard that included wood shavings and paper shred bedding. Animal body weight and activity were tracked every two days. Special training of the first author was provided in animal handling, anesthesia, and intravenous injection. All efforts were made to minimize the suffering of the animals. Genomic DNA was purified from tail biopsies by isopropanol precipitation, and the transgene was amplified by PCR using the forward primer AGGACTGACCACTCGACCAG and the reverse primer CGGGGGTCTAGTTCT GCAT. Resulting PCR products of 377 base pairs (bp) were analyzed by 2% agarose gel electrophoresis.

### 4.7. Small Animal PET/CT Imaging Experimental Procedures

Radiosynthesis and formulation of [^18^F]TFAHA were performed as previously described [21]. Twenty minutes prior to imaging, each mouse was injected intravenously with 8.04 ± 0.75 MBq/0.1 mL of [^18^F]TFAHA for assessment of epigenetics or 39.5 ± 3.5 MBq/0.2 mL of [^11^C]Pittsburgh compound-B (PiB) for detection of Aβ deposition. Mice were then anesthetized with 1–1.5% isoflurane in 100% O_2_ through a nose cone for static imaging by the Triumph preclinical PET/SPECT/CT system (Gamma Medica-Ideas, Northridge, CA, USA). PET data were acquired for 20 min and then reconstructed with a filtered background projection probability algorithm, and an additional CT scan was performed for anatomical localization. Subsequently, PET and CT images were co-registered by PMOD 3.5 software package (Pmod Technologies, Zürich, Switzerland). The uptake and regional retention of these radiotracers were processed and analyzed. The values were reported as standardized uptake values (SUV) representing the mean activity values for each whole brain or the regional uptake normalized to the injected dose per body weight of each individual animal.

### 4.8. Immunofluorescence (IF) Staining

Cryosections were fixed in 4% paraformaldehyde, washed with PBS, and permeabilized with 0.025% Triton X-100 twice for 5 min before blotting with 10% normal goat serum and 1% BSA for 2 h at room temperature. Then, cryosections or cell culture slides were incubated with the primary antibodies overnight at 4 °C after washing with PBS twice and fixing with 4% formaldehyde. Then, the slides were incubated with Alexa 488-, Alexa 594-, or Cy5-conjugated secondary antibodies (Abcam, Cambridge, UK) and DAPI for 1 h in the dark at room temperature. All slides were then mounted in ProLong Antifade Mounting Medium (catalog #P36970, Life Technologies, Carlsbad, CA, USA), and coverslips were applied before visualization under a confocal fluorescence microscopy. Five images from random fields were obtained and analyzed using a laser-scanning confocal microscope (Zeiss LSM 880, Jena, Germany) with Zen Blue software (ZEISS, Jena, Germany) or a fluorescence microscope (Zeiss AX10, Jena, Germany). Fluorescent intensities and cell numbers were quantified by ImageJ (NIH). The following primary antibodies were used: polyclonal anti-HDAC1 (#ab19845, Abcam, Cambridge, UK), anti-HDAC4 (#ab79521, Abcam, Cambridge, UK), anti-HDAC5 (#ab55403, Abcam, Cambridge, UK) and monoclonal anti-amyloid-β antibody (Sig-39220, Cell signaling, Danvers, MA, USA).

### 4.9. Statistical Analysis

Quantitative results were expressed as mean ± SEM. Data were subjected to one-way analysis of variance (ANOVA) followed by Student’s *t*-test, as appropriate, with GraphPad Prism v.9.0 software (GraphPad, San Diego, CA, USA). A *p* value less than 0.05 was considered statistically significant.

## 5. Conclusions

This study demonstrated that the expression of class IIa HDACs, mainly HDAC4, in neuronal cells was responsive to the exposure to neurotoxic Aβs in a dose-dependent manner. Inhibition of HDAC4 by the selective inhibitor TasQ partly rescued the expression of genes related to neuronal memory/synaptic plasticity and showed the effect of an HDAC4 targeting treatment. Furthermore, PET imaging with [^18^F]TFAHA provided a quantitative in vivo evaluation of class IIa HDACs expression in the brain of AD transgenic mice. These results highlight the importance of epigenetic regulation in AD and further encourage the development of neuro-epigenetic imaging approaches and therapies.

## Figures and Tables

**Figure 1 ijms-22-08633-f001:**
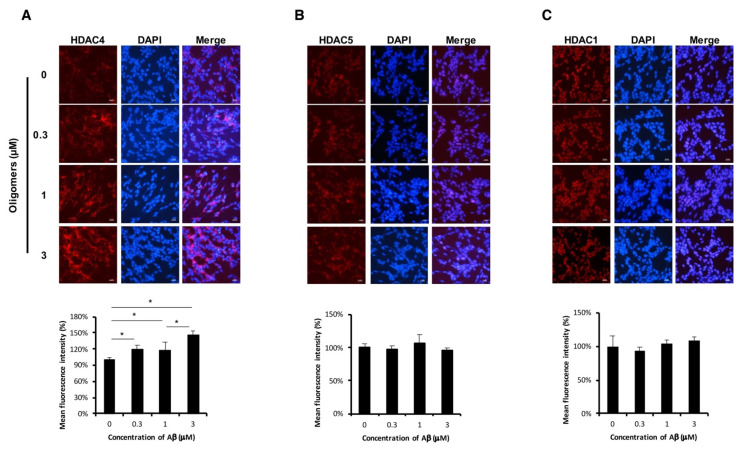
Representative images of (**A**) HDAC4, (**B**) HDAC5, and (**C**) HDAC1 expression in the neuroblastoma cell line SH-SY5Y after 48 h of exposure to an increasing concentration of Aβ oligomers. Scale bar, 2 μm. Bottom graphs depict the quantification of immunofluorescent intensity of target proteins in Aβ oligomers-treated cells, relative to the untreated control. Data are mean ± SEM. One-way ANOVA was used (F = 9.54, *p* = 0.0021. * *p* < 0.05 by Tukey’s post hoc test).

**Figure 2 ijms-22-08633-f002:**
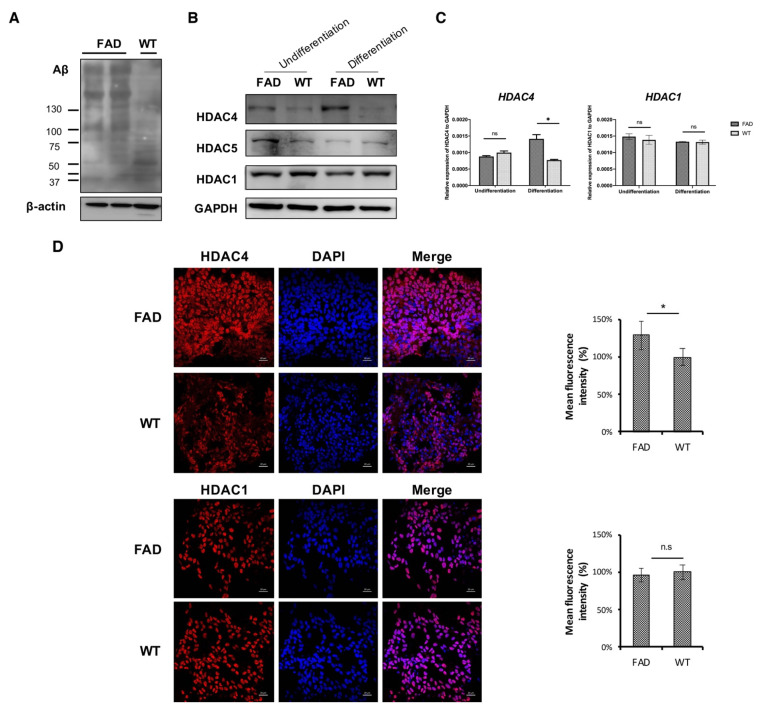
The level of HDAC4 was increased in the FAD human neural cell model. (**A**) Aβ expression was confirmed by western blot. Internal control: β-actin. (**B**) Representative Western blot of HDAC4, HDAC5, and HDAC1 on day 7 after the induction of differentiation. Internal control: GAPDH. (**C**) Quantitative RT-PCR analysis of *HDAC4* (left panel) and *HDAC1* (right panel) in differentiated FAD cells. Gene expression levels were normalized against GAPDH levels in each sample. Data are mean ± SD. * *p* < 0.05, and n.s., nonsignificant by Student’s *t* test. * *p* = 0.0117, FAD versus WT. (**D**) Representative confocal micrographs of FAD differentiated cells immunolabeled with HDAC4. Right graphs depict the quantification of HDAC4 labeling intensity in differentiated FAD cells relative to WT control. * *p* = 0.0295, FAD versus WT by Student’s *t* test.

**Figure 3 ijms-22-08633-f003:**
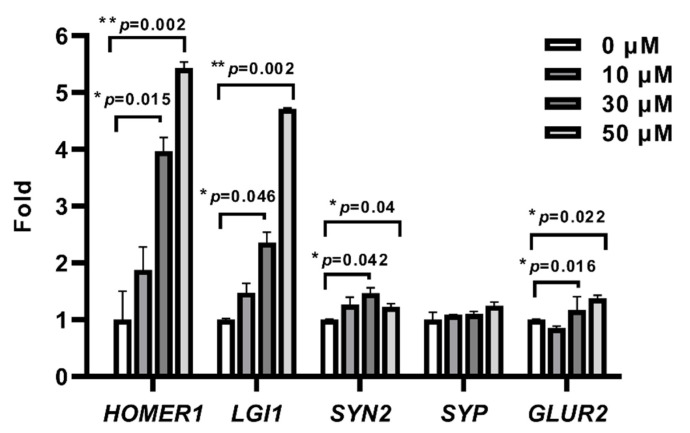
Quantitative RT-PCR analysis of neuron and plasticity-related genes in differentiated FAD cells after 48 h exposure to an increasing concentration of Tasquinimod. Data are expressed as mean ± SEM. * *p* < 0.05, ** *p* < 0.01 by Student’ *t*-test. Significant increase relative to 0 µM (vehicle control). Abbreviation: *HOMER1*, homer protein homolog 1; *LGI1*, leucine-rich glioma inactivated 1; *SYN2*, synapsin II; *SYP*, synaptophysin; *GLUR2*, glutamate receptor 2.

**Figure 4 ijms-22-08633-f004:**
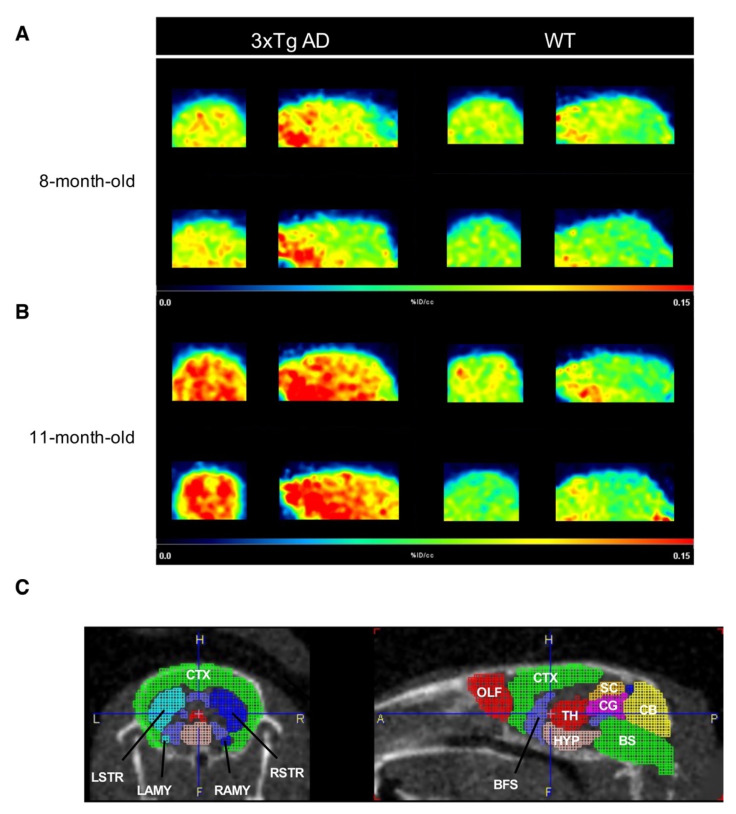
In vivo epigenetic imaging of using [^18^F]TFAHA. Representative [^18^F]TFAHA PET imaging of 3xTg AD mice and WT mice at (**A**) 8 and (**B**) 11 months of age. Coronal (left) and sagittal (right) slices were projected on a built-in T1 MRI mouse template of Pmod software (scale by %ID/c.c). (**C**) Schematic illustration of a mouse coronal and sagittal brain section depicting several regions of interest (ROI) used for the quantification of PET imaging. Abbreviation: CTX, cortex; STR, striatum; TH, thalamus; BFS, basal forebrain; AMY, amygdala; CB, cerebellum; OLF, olfactory bulb; CG, cingulate cortex; BS, pons plus medulla.

**Figure 5 ijms-22-08633-f005:**
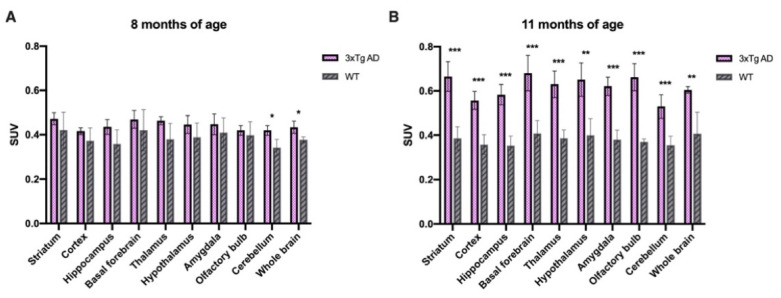
The uptake of [^18^F]TFAHA was quantified and expressed in terms of standard uptake values (SUV) in different brain regions of 3xTg AD and WT mice at (**A**) 8 and (**B**) 11 months of age. Data are expressed as mean ± SD (n = 4–5/each group). The uptake in the whole brain of 3xTg AD versus WT mice was 0.433 ± 0.028 versus 0.377 ± 0.013 at 8 months of age, *p* = 0.011, and 0.604 ± 0.015 versus 0.371 ± 0.013 at 11 months of age, *p* = 0.0056 by Student’s *t*-test, * *p* < 0.05, ** *p* < 0.01, *** *p* < 0.001 by Student’s *t* test.

## Data Availability

The data presented in this study are available in the article and Appendix A.

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
