# Peer review of "Evaluation of Class IIa Histone Deacetylases Expression and In Vivo Epigenetic Imaging in a Transgenic Mouse Model of Alzheimer’s Disease"

_ijms, 2021, doi:10.3390/ijms22168633_

Round 1
Reviewer 1 Report
The tracing of class iia hdacs in vivo could be of interest in consideration of the reported role of hdac4 in AD. However many data are missing and also the pharmacological approach is quite weak.
- 18F TFAha PET tracing: confirm the targeting of class ìia hdacs by comparative approaches (and maybe the use of ko mice)
- Tasquinimod is not specific, compare with TMP195 TMP269 LMK235
- IF in fig 1 shows pan/cytoplasmic localization of Hdac4, ahile it is nuclear in the following IF. Explain and use leptomycin as a control
- Tasquinimod is predicted to bind the c terminus, while the glut rich region of hdac4 is at n term, explain
- Which is the effect of Tasquinimod and TMP or LMK235 on amyloid fiber formation
- Why hdac5 is not regulated during AD like hdac4. Provide any details about the mechanism
Reviewer 2 Report
The authors did a good job with imaging class IIa HDAC4 in mouse model for alzheimer's disease. I would recommend the article for publication after the authors address the points below:
- Figure 1A: It is difficult to tell if the expression level of HDAC4 is affected by increasing concentration of oligomers. One more data point at a concentration higher than 3µM will help. Also, choice of color for the HDAC panel makes it difficult to see increasing level of expression. I would suggest using red color as the authors did in Figure 2D.
- Align Figure legend with the starting end of Figures 1& 2.
- In Line 176, the authors stated that [18F]TFAHA had highest concentration in the heart after iv administration. I was wondering if this was seen in both 3xTg AD mice and wild type mice. Was there any sign of toxicity to the mice? Also What is the half-life of [18F]TFAHA in the mice body (especially in the heart)? The authors should address this in their discussion, and maybe speculate on the potential toxic side effect of using this radiotracer since it also accumulate abundantly in other organs.
- In Line 266, in addition to the radiotracers listed by the authors, [18F]Bavarostat (check https://doi.org/10.1021/acscentsci.7b00274) is another well characterized radiotracer used to visualize HDAC6 in the brain.
- Line 297, “subsequently” should be “subsequent”
Reviewer 3 Report
In this article, authors studied the role of class IIA HDAC expression in Alzheimer’s disease model and monitor in vivo expression of HDAC4 using [18F]TFAHA-PET imaging. Conclusions are over-stated without any solid evidence.
There is not enough detail the material methods how authors established FAD model. In their FAD model, HDAC5 levels decreases upon differentiation (Figure 2B), contrary to HDAC4. It seems that authors ignored this fact and just showed the data on HDAC1 and HDAC4. It also means this is not specific to HDAC4.
Authors checked the expression of few memory related genes, but did not check the histone acetylation levels on the promoter of these genes in their FAD model and also did not show the binding of HDAC4 on these genes to show if they are being regulated by HDACS, specially by HDAC4.
Figure 3- authors should have shown control with HDAC inhibitors for the expression of memory related genes. By the way, there are many memory related genes and I did not get the logic why only these genes were selected? Also gene name should be italic and authors should fix this in the whole manuscript.
Figure4- It is hard to orientate ourselves to find specific brain regions. At 8 months, whole brain increase is due to cerebral?
Figure 2SB- no scale bar.
Round 2
Reviewer 1 Report
The authors have addressed the points raised by me only theoretically (with the exception of Leptomycin B treatment).The validation of the technique on Hdacs Ko mice/injected cells or by using specific inhibitors rather than Tasquinimod (whose specificity must be demonstrated,while the specificity of MC1568 is questioned by many authors) is in my opinion a central point that must be sustained experimentally.
Reviewer 3 Report
Authors answered all the queries.
Author Response
Once again, thank you very much for your comments and suggestions.
Round 3
Reviewer 1 Report
Figure 2b and c: explain why Hdac4 RNA is downregulated in undifferentiated FAD cells in respect to Wt while the protein is upregulated.
New s7 about RNA espression levels and the effect of Hdac inhibitors. Place the statistics for all the comparisons. In the text spell correctly MC1568. I do not understand the rational to use 10uM of the inhibitors. Moreover, did the cells survive at 10uM SAHA treatment?
MC1568 is not a class IIa inhibitor, strenght this in the discussion. The data obtained with 100uM Tasquinimod are also very weak as this high concentration do not support any specificity. This must be discussed in the paper.
